# Multifunctional Optical Device with a Continuous Tunability over 500 nm Spectral Range Using Polymerized Cholesteric Liquid Crystals

**DOI:** 10.3390/polym13213720

**Published:** 2021-10-28

**Authors:** Mi-Yun Jeong, Hyeon-Jong Choi, Keumcheol Kwak, Younghun Yu

**Affiliations:** 1Department of Physics and Research Institute of Natural Science, Gyeongsang National University, Jinju-si 52828, Gyeongsangnam-do, Korea; choihj@gnu.ac.kr; 2School of Electronics and Electrical Engineering, Sungkyunkwan University, Jangan-Gu, Suwon 16419, Gyeonggi-do, Korea; kkwak95@skku.edu; 3Department of Physics, Jeju National University, Jeju-si 690756, Jeju-do, Korea; yyhyoung@jejunu.ac.kr

**Keywords:** optical multifunctional device, optical tunable filter, liquid crystal filter device, circular polarizer, intensity-variable beam splitter

## Abstract

We report that polymerization makes a robust, practically applicable multifunctional optical device with a continuous wavelength tunable over 500 nm spectral range using UV-polymerizable cholesteric liquid crystals (CLCs). It can be used as a circular polarizer generating an extremely high degree of circularly polarized light with |g| = 1.85~2.00. It can also be used for optical notch filters, bandwidth-variable (from ~28 nm to ~93 nm) bandpass filters, mirrors, and intensity-variable beam splitters. Furthermore, this CLC device shows excellent stability owing to the polymerization of CLC cells. Its performance remains constant for a long time (~2 years) after a high-temperature exposure (170 °C for 1 h) and an extremely high laser beam intensity exposure (~143 W/cm^2^ of CW 532 nm diode laser and ~2.98 MW/cm^2^ of Nd: YAG pulse laser operation for two hours, respectively). The optical properties of polymerized CLC were theoretically analyzed by Berreman’s 4 × 4 matrix method. The characteristics of this device were significantly improved by introducing an anti-reflection layer on the device. This wavelength-tunable and multifunctional device could dramatically increase optical research efficiency in various spectroscopic works. It could be applied to many instruments using visible and near-infrared wavelengths.

## 1. Introduction

Basic optics such as mirrors, circular polarizers, filters, and beam splitters are essential elements in optics and photonics. For decades, these devices have been developed from various materials such as dielectric, crystalline, metallic, and metamaterials. They have been developed with advanced technologies depending on the wavelength and the intensity of light available [1,2,3,4].

Liquid crystals have been widely applied as a display material throughout the world. Cholesteric liquid crystals (CLCs) are a liquid crystal phase characteristic of photonic crystals [5,6,7,8]. Since they are an important material for optical elements, e-books, and reflective displays, various studies and developments are underway [5,6,7,8,9,10,11,12]. In particular, spiral nanostructures of CLCs can be used as laser resonators due to their selective reflective properties, thus drawing keen attention from laboratories and academia [5,6,7,8,13,14,15]. Recently, the applicability of CLCs as another wavelength tuning and multifunction optical component other than lasers or displays has been reported [16,17]. Not long ago, we developed optical filter devices using CLCs, reported their optical versatility function, and showed their applicability as optical devices in two papers [16,17]. In the first paper [16], CLC cell devices showed the function of a continuously wavelength-tunable optical notch filter over a 100 nm spectral range by rotating them. They had the functions of a mirror and a beam splitter. They also had a stable operation under a high laser intensity (~124 W/cm^2^ of CW 532 nm diode laser and ~4.43 MW/cm^2^ of Nd: YAG pulse laser operation for 2 h) without showing damage. However, general CLCs are sensitive to temperature changes. Thus, the photonic bandgap (PBG) wavelength position can move when ambient temperature changes, thus requiring the selection of liquid crystals not affected by temperature changes. In the second paper [17], CLC devices with spatial pitch gradients by combining four CLC cells showed continuously tunable and bandwidth-variable optical notch and bandpass filters. The band wavelength position could be spatially tuned from 470 nm to 1000 nm, and the bandwidth could be reversibly controlled from the original bandwidth (from 60 nm to 18 nm). Despite the superior performance of such CLC devices, CLC cells were made by general CLCs that are not polymerized. Thus, serious problems remain to be solved to utilize them as devices. Due to general CLCs, a well-developed pitch gradient of the wedge CLC cell, which is an essential condition for a wide range of continuous wavelength tunings of the photonic bandgap (PBG), does not have long-term stability. The thermodynamic diffusion of chiral molecules continuously happens in general CLC cells with a chiral molecule concentration gradient. Spatially developed pitch gradients in wedge cells will eventually disappear in months [18]. Stability against temporal and thermal stimuli of helical pitch gradients is a prerequisite for applying CLC devices as practical devices. Thus, the primary purpose of this study was to solve the stability problem of CLC devices by introducing polymerized CLC structures. We developed a polymerized CLC (PCLC) device system with optical multifunction of a left- or right-circular polarizer generating an extremely high degree of circularly polarized light with a dissymmetry factor |g| = 1.98, notch or bandpass filter, mirror, and beam splitter that could be utilized in practical devices. After forming a continuous spatial pitch gradient in the wedge cell over 500 nm spectral range (from ~450 nm to ~950 nm), the CLC structure was polymerized by UV. By fabricating the device with a polymerizable CLC, we could solve the unstable problem of pitch gradient in the general CLC structure shown in our previous study (the second paper mentioned earlier) [17]. The strategy of forming a continuous spatial pitch gradient in the wedge CLC cell can increase the wavelength tuning range by five times compared to the results of our previous study (the first paper mentioned earlier) [16]. Polymerized CLC cells also showed excellent stability performance after a long time (about two years), a high-temperature exposure (170 °C for 1 h), and an extremely high laser beam intensity exposure (~143 W/cm^2^ CW 532 nm CW diode laser and 2.98 MW/cm^2^ Nd: YAG pulse for 2 h each). Their optical properties were studied theoretically, including the refractive index, pitch, and FWHM of the photonic bandgap and the thickness of the polymerized CLC cell by Berreman’s 4 × 4 matrix method. By introducing an anti-reflection layer on the device, the characteristics of the device were also significantly improved. As far as we know, such a wavelength-tunable multifunctional device with a high performance has not been reported yet. Such a device could dramatically increase optical research efficiency in various spectrometer tasks in the future. It could be applied to many devices using visible (VIS) and near-infrared (NIR) wavelengths.

## 2. Optical Properties of CLC Layers with Helical Nanostructures

CLCs are a mixture of nematic liquid crystals (NLCs) and chiral molecules. They form self-organized periodic helical nanostructures in a cell with an alignment layer. An important intrinsic optical characteristic of the CLC structure is its circular polarization of incident light. Figure 1 shows a fundamental optical property of the CLC cell. For an unpolarized incident light, a CLC cell with a right-handed helix reflects right-circularly polarized light and transmits left-circularly polarized light in a photonic bandgap wavelength range. Therefore, for a light with the same handedness as the CLC, selective (Bragg) reflection occurs in a wavelength range $n_{o}\times p<\lambda <n_{e}\times p$ with a photonic bandgap (PBG) (|$n_{e}\;$ − $\;n_{o}$|× p), where p (pitch) is the length for one full rotation of the director around the helix axis (it could be controlled by the ratio of chiral molecules to the nematic liquid crystal) and $n_{o}$ and $n_{e}\;$ are ordinary and extraordinary refractive indices of nematic molecules, respectively [15,16,17,18,19,20]. We can obtain left- and right-circularly polarized light in the selective band of a CLC cell. Therefore, CLCs work as a circular polarizer in the photonic bandgap wavelength range. Figure 1 also shows the principle of a notch filter by a polymerized CLC with a right-handed helix (R-PCLC) and a polymerized CLC with a left-handed helix (L-PCLC). For unpolarized incident light, right-circularly polarized light in the band is reflected at the CLC with a right-handed helix, and left-circularly polarized light in the band is reflected at the CLC with left-handed helix, in turn. Furthermore, combining some L-PCLC cells and R-PCLC cells could be transformed into other optical devices such as a bandpass filter, a beam splitter, and a mirror. We can see the multifunctional properties of PCLC cells one by one in the following sections.

Another primary advantage of this device is its continuous wavelength variability in the wideband wavelength range.

To realize the continuous wavelength tunability of circular polarization in a broad spectral range over 500 nm, the use of a wedge PCLC cell structure and a continuous pitch formation in the wedge direction of the cell could be one solution (see Figure 2). As a critical and required condition, the pitch gradient of PCLCs must match the spiral pitch change determined by cell thickness change in the wedge direction of the PCLC cell [18]. Figure 2 shows a schematic diagram of the PCLC structure corresponding to the ideal PCLC wedge cell with a continuous pitch gradient. Cholesteric helical pitches are quantized with the number of half-turns [13,19] to satisfy the boundary condition of the wedge CLC cell with an alignment layer. Along the wedge direction (+x-direction in Figure 2), a linear increase in the helical pitch (*p*) continuously occurred. Therefore, spatially, the wavelength position of the PBG ($n_{o}\times p<\lambda <n_{e}\times p$) also continuously changed according to the pitch change along the wedge direction on the PCLC cell. A method to fabricate the PCLC wedge cell structure with a pitch gradient over 500 nm spectral range is shown in Section 3 (Fabrication of PCLC cells).

## 3. Fabrication of PCLC Cells

To make polymerized CLC cells (PCLC cells) that have a fine spatial pitch gradient in a vast wavelength range of ~500 nm, a chiral molecular concentration gradient was formed along the wedge direction in the cell, as shown in Figure 2. In our previous study, introducing a BK7 glass substrate coated with an anti-reflective layer (AR layer) on one side improved the refractive index mismatch between the air and glass layer of the CLC cell [16,17]. As a result, the transmission characteristics of filter devices were greatly enhanced. Therefore, we used the same BK7 plate with an AR layer for a spectral range from 400 nm to 1000 nm (0.6 μm in thickness, Yunam Optics, Bucheon, Gyeonggi, Korea) in this study to obtain a good transmission efficiency [16,17]. On the BK7 surface of each plate, a layer of polyimide (PI) (7492K, Nissan Chem Korea Co., Ltd., Seoul, Korea) was spin-coated and then heat-treated to crosslink PI. A PI/BK7/AR layer was fabricated by rubbing the PI layer with a rubbing cloth (HC20, cotton, NES TECHNOLOGY Co., Ltd., Gyeonggi, Korea) to make the PI an alignment layer. With the PI-coated surface facing inward, empty wedge cells were fabricated with two plates employing spacers of two different sizes (thin and thick, ~25 µm and ~30 µm in thickness, respectively) with a lateral distance of ~1.8 cm. We fabricated two kinds of polymerized CLC (PCLC) wedge cells with a uniform helical pitch gradient using UV-curable CLCs. The pitch gradient of CLC cells was made using two (high and low) chiral dopant concentrations of CLCs. For R-PCLC cells with right-handed helicity, RMS11-066 (with a center wavelength of the photonic bandgap of λ_B_ = ~400 nm) was used as a high chiral dopant concentration CLC. A mixture of ~0.105 g of RMS11-066 without solvent and 0.200 g of RM141C (with λ_B_ = ~1000 nm, both from Merck) was used for low chiral dopant concentration CLCs. For L-PCLC cells with left-handed helicity, RM-xx (with λ_B_ = ~400 nm) was used as a high chiral dopant concentration CLC, and a mixture of ~0.105 g of RM-xx without solvent and 0.200 g of RM141C (with λ_B_ = ~1000 nm, Merck) was used as a low chiral dopant concentration CLC. Initially, these CLCs were solutions of reactive mesogens in toluene and contained photoinitiators. The solvent in the CLC solution was evaporated entirely before use. The previously made empty cells were half-filled with a high (low) chiral dopant concentration CLC at a thin (thick) spacer position. These CLC cells were then maintained in a dark oven at ~60 °C for about four weeks to develop a chiral dopant concentration gradient in the wedge direction of the cell through the diffusion of helical rotary powers in the CLC. Subsequently, CLC cells with a uniform helical pitch gradient were polymerized after exposure to UV radiation (25 mW/cm^2^ with 365 nm wavelength) for about 3 h.

Figure 3a shows a photograph of four wedge cells with well-formed pitch gradients (R-PCLC1, R-PCLC2, L-PCLC1, and L-PCLC2 cells) on a plotting paper. Figure 3b is a juxtaposition of ten polarized microscope images at different spatial positions of the R-PCLC1 cell. A continuous color change stemming from a continuous pitch change in a vast wavelength range of ~500 nm along the wedge direction of the cell was observed. We can also identify well-developed pitch changes with continuous wavelength changes in the PBG in the next section.

## 4. Experimental Results and Discussion

Figure 4 shows the experimental setup containing two different paths (1 and 2). Path 1 was used to verify the functionality of notched filters and circular polarizers in R-PCLC and L-PCLC cells. Path 2 was used to check the function of a bandpass filter. When Path 1 was selected, the Ag mirror (M; Average reflectance > 98%) in Path 1 had to be removed. Depending on the purpose of the device, the number of PCLC cells was determined. Each PCLC cell was located above the rotator, which was placed on a translation stage. Transmitted data of PCLC cells were collected independently by moving each translation stage in a lateral direction or by varying the angle of incidence using each rotator. Light signals from PCLC cells were acquired through waveguides (W1 or W2) with a spectrometer (OOI, USB2000+).

### 4.1. Tunable Multi-Functionality over 500 nm Wavelength: Circular Polarizer, Notch, Bandpass Filter, Mirror, and Intensity-Variable Beam Splitter

Figure 5 shows transmitted PBG spectra and polar plots of circularly polarized light intensities from the polymerized R-PCLC1 cell for vertically incident beams using the translational stage T1. All circularly polarized light intensity data are normalized to mean values. The R-PCLC1 was thermally treated at 170 °C for 1 h after its fabrication and aged for more than two years. Considering the principle of action of CLCs by linking results of Figure 5 to Figure 1 and Figure 2, only one L-PCLC (or R-PCLC) cell could be used to obtain circularly polarized light in a wide wavelength range of 450 nm to 960 nm. We compared the degree of circular polarization of circularly polarized light transmitted through the PCLC cell with that of circular polarization by the λ4 plate. We used a HeNe laser (632.8 nm) (Figure 5b) and a halogen and deuterium lamp (OOI system). In addition, the circularly polarized light passing through the R-PCLC1 cell was again linearly polarized using the  λ4 plate to confirm the quality of polarization of the circularly polarized light. Additionally, we calculated the degree of circular polarizations g, which is defined as g=|2(IL−IR)(IL+IR)|, where $I_{L}$ and $I_{R}$ are denoted intensities of left- and right-handed circularly polarized light, respectively. The circularly polarized light data at 450 nm, 633 nm, 780 nm, and 900 nm are fitted by a rotated elliptic equation, $\frac{{\left(xcos\left(\phi \right)+ysin\left(\phi \right)\right)}^{2}}{a^{2}}$ + $\;\frac{{\left(-xsin\left(\phi \right)+ycos\left(\phi \right)\right)}^{2}}{b^{2}}=1$, where a is the length of the semi-major axis, b is the length of the semi-minor axis, ϕ is the rotation angle of the major axis, and $r=\sqrt{x^{2}+y^{2}}$ . Figure 5a shows continuous PBG transmission spectra of the R-PCLC1 cell made by moving laterally for vertically incident beams in Path 1 of Figure 4.

Figure 5b shows the degree of circular polarization of light [21,22] passing through the PBG with a central wavelength of 633 nm for the R-PCLC1 cell by using a 632.8 nm HeNe laser (Melles Griot, beam diameter = 700 μm). When a linearly polarized 632.8 nm laser light passes through at a location in the R-PCLC1 cell with a PBG center wavelength at ~633 nm, the right-circularly polarized light reflects, and the left-circularly polarized light is transmitted. The transmitted light passes through a linear polarizer (analyzer) and a pinhole (600 μm) to check the circular polarization state. The pinhole is used for excluding the light that passes outside of the 632.8 nm PBG region. The data (□) show a polar plot (r,θ) of left-circularly polarized light made by a linear polarizer and a λ4 plate of 632.8 nm; the data (o) show a left-circularly polarized light passed through the R-PCLC1 cell with a PBG center wavelength at 633 nm; and the data (△) show a linearly polarized light intensity that is converted from the left-circularly polarized light passed through the R-PCLC1 cell by a λ4 plate (632.8 nm).

Polar plots are obtained by using the halogen and deuterium lamp (OOI system); the left-circularly polarized lights passed through the R-PCLC1 cell with a PBG center wavelength at 450 nm (c), 633 nm (d), 780 nm (e), and 900 nm (f), respectively. Each solid red line is fitting curves by the elliptic equation. The data (△) in (d) are a linearly polarized light intensity that is converted from the data (o) of (d) by a λ4 plate (632.8 nm). The length of semi-major axis a, the length of semi-minor axis b, and the ellipticity (b/a) of the ellipse obtained from the fitting data are given in Table 1.

Figure 5b (△) shows that the light is entirely linearly polarized—maximum intensity ($I_{L})$ is 1.033 at the angles 0° and 180°, and minimum intensity $\left(I_{R\;}\right)$ is 0 at the angles 90° and 270° of the analyzer. This means that the transmitted light passed through the R-PCLC1 is a complete left-circularly polarized light. The g value of the R-PCLC1 at 632.8 nm by the HeNe laser is calculated as 2, and where $I_{L}=1\;\left(max.value\right)\;\mathrm{and}\;I_{R\;}=0\;\left(min.\;value\right).$ Additionally, Figure 5d (△) shows that the maximum intensities ($I_{L})$ are 1.002 at 85° and 0.930 at 265°, and the minimum intensities $\left(I_{R\;}\right)$ are 0.037 at 175° and 355° of the analyzer. The g value of the R-PCLC1 at 633 nm by the lamp is calculated as 1.857~1.847. It can be seen that the g value by the polarized laser light is much better than the g value using unpolarized lamplight. In the case of lamp light sources, it seems that there are many wavelengths involved, and the degree of collimation of light is relatively low. In both cases, the g value is very high, indicating that the degree of circular polarization passing through the R-PCLC1 cell is excellent. Additionally, from the results in Figure 5 and the ellipticities (b/a) in Table 1, it can be seen that the degree of circular polarization by the R-PCLC1 cell is slightly smaller than that of the λ4 plate. In the case of 450 nm in Figure 5c, it seems that the degree of circular polarization appears to be relatively low due to the reflection by the mismatch in the refractive index of the boundary or absorption effect.

On the other hand, we could see the superiority of our device by comparing the existing method to the method of using our device in creating circular polarization: to make a circularly polarized light, a linear polarizer and a λ4 plate corresponding to an incident wavelength (λ) are required. Circular polarization of 100 wavelengths requires 100 different λ4 plates. However, when using an R-PCLC (or L-PCLC) cell to produce left (or right) circularly polarized light, it only needs to pass through the R-PCLC (or L-PCLC) cell.

Figure 6 shows continuous tuning PBG spectra of CLC notch filter systems with pitch gradients.

To change the notch wavelength position of CLC cells, CLC cells were laterally moved (Figure 6a,b) or rotated (Figure 6c,d). Wavelength tuning notch data were obtained via Path 1 of Figure 4. Figure 6a shows notch spectra from ~430 nm to 940 nm by lateral movement depending on the spatial position of the continuous tunable notch filter system using a set of one notch systems consisting of an R-PCLC1 cell placed in the translation stage T1 and an L-PCLC1 cell placed in the translation stage T2. Data were collected by simultaneously moving T1 and T2 to maintain the same PBG position for both cells. For unpolarized incident light, the right-circularly polarized light in a photonic band was reflected at the R-PCLC1 cell, and the left-circularly polarized light in the same photonic band to the R-PCLC1 cell was reflected at the L-PCLC1 cell, in turn. By moving to the lateral thick wedge direction in the cell with a pitch gradient, the FWHM of the PBG gradually increased from ~50 nm at 440 nm to ~93 nm at 964 nm. This broad optical bandgap width resulted from the intrinsic refractive index of the material because the bandgap was determined by its ordinary (*n_o_*) and extraordinary (*n_e_*) refractive index in the form of |*n_e_* − *n_o_*| × *p*, where *p* was a pitch. Using a CLC material with small Δ*n* (=*n_e_* − *n_o_*) would be helpful to reduce the bandgap width.

Theoretically, all light within the photonic band is reflected, and light outside the band is transmitted, as shown in Figure 1. However, there was light leakage of up to ~5% across the spectral range within the band, although the AR layer between the air and BK7 improved the reflective index mismatch. It seemed that the refractive index difference at the PI layer/PCLC layer and the PI layer/BK7 boundaries of the CLC cells caused this light leakage. Figure 6b shows notch spectra from 440 nm to 950 nm using a multiple cell reflection method with two sets of notch filter systems consisting of two R-PCLC cells and two L-PCLC cells. Although multi-cell reflection methods have been applied to improve light leakage problems in the band, some light leaks still exist within the band. When polymerized CLCs were used, there was a ~1.6% light leak in the 500 nm to 800 nm wavelength. This was mainly estimated to be due to the discrepancy in the refractive index between the PCLC layer and the PI layer by comparing with the case of using general CLC cells where light leakage was not visible in the band [16,17]. When measuring the spectrum, each cell’s PBG positions were adjusted to maximize the wavelength range with a flat minimum value in the notch band. Therefore, the FWHM of the PBG of the four-CLC-cell system is more prominent than in the two-CLC-cell system.

Figure 6c shows the notch spectrum from ~430 nm to ~940 nm by rotation using a set of notch systems consisting of an R-PCLC1 cell placed on the rotator R1 and an L-PCLC1 cell placed on the rotator R2. Placing cells at a distance of ~2.6 cm from the rotation axis could change the wavelength in the broader wavelength area as cells rotated. This was due to the simultaneous effect of the PBG moving to different locations and increasing the angle of incidence for CLC cells. Data were collected by rotating R1 and R2 between –20 and +20 degrees while maintaining the same PBG position for both cells. There was some light leakage of up to ~6% within the band across the spectral range. Figure 6d shows notch spectra from 440 nm to 950 nm using the multiple cell reflection method by rotation with two sets of notch filters. Using two sets of notches, when rotating the cells to collect data, the light leakage within the band was 2~3% due to the discrepancy in the refractive index between the PCLC layer and PI layer (Figure 6d as in the Figure 6b result). In the results of Figure 6a,c, employing one notch filter set, the transmittance in the outer region of the band was greater than 90% in the wavelength region of more than 600 nm. In the wavelength region from 600 nm to 400 nm, the band’s transmittance of the outer region decreased from ~90% to 40%. In Figure 6b,d, with two notch filter sets, the transmittance in the outer region of the band was ~90% in the wavelength region of more than 600 nm. When the wavelength decreased from 600 nm to 400 nm, the band’s transmittance of the outer region decreased from ~90% to 20%, which seemed to be mainly due to absorption and refractive index mismatch at boundaries of the PCLC layer and the PI layer. From the Figure 6b,d results, in order to experimentally examine the effect of the PI layer on the serious decrease in transmittance from 600 nm to 400 nm, the reflectance and transmittance of the substrate used to manufacture the PCLC cell were investigated. Figure 6e shows the reflectance of the BK7 plate, AR/BK7 plate, and AR/BK7/PI plate. Additionally, Figure 6f shows the transmittance of the BK7 plate, the AR/BK7 plate, and the AR/BK7/PI plate, respectively. In the case of AR/BK7 substrates, AR coating on one side of the BK7 substrate reduces (increases) reflectance (transmittance) uniformly in the wavelength region of 400 nm to 1000 nm. However, for the AR/BK7/PI plate, the reflectance (transmittance) rate increases (decreases) again, and this effect increases, especially as it moves from the 600 nm to 400 nm region. This seems to be the effect of refractive index mismatch and absorption at the interface between the BK7/PI layers in this wavelength region. Therefore, as the number of PCLC cells used increases, the number of substrates increases, and the transmission results of Figure 6b,d appear, and this part needs to be improved further in the future.

Figure 7 shows continuous bandpass filter spectra from ~380 nm to ~920 nm from the tunable bandpass filter system consisting of a mirror, the R-PCLC1 (S_1′_), and the L-PCLC1 (S_2′_) in Path 2 of Figure 4. Data were measured by laterally moving each R-PCLC1 and the L-PCLC1 cells with translation stages of T_5_ and T_6_ to maintain the same PBG position. For an unpolarized incident light, the right-circularly polarized light in the band of R-PCLC1 and left-circularly polarized light in the same band of L-PCLC1 were reflected, respectively (see Figure 1). Thus, all light in the band was reflected. As all light in the band was reflected, this bandpass filter system was polarization-independent. The transmittance of the band was as high as 90–100% in the 500~950 nm range. As it moved from 500 nm to 400 nm, the transmittance of the band was decreased from 90% to 60%. From the results of Figure 6 and Figure 7, we could also say that this transmittance decrease was due to absorption and refractive index mismatch at boundaries of the PCLC layer and PI layer. In the PBG spectra position of Figure 7, there was a blue shift across the entire spectrum region compared to those of Figure 6. The reason was that the angle of the incident beam was rotated 20 degrees to cells in Path 2 of Figure 4. Thus, the band position of the PBG was blue-shifted by ~50 nm [16]. Furthermore, the bandpass filter with more than 90% reflection at high laser power could also be used as a mirror.

The FWHM of the PBG spectrum of the fabricated samples is so broad from ~50 nm to ~93 nm (Figure 8a). This behavior comes from the refractive index of CLC materials. However, this intrinsic behavior of bandwidth could be controlled by combining two R-CLC cells (or two L-CLC cells). By putting R-PCLC1 in the mirror (M) position and putting R-PCLC2 in the S_1′_ place in Path 2 of Figure 4, the bandpass filter bandwidth can be actively controlled. Figure 8b shows reduced bandwidth spectra from ~93 nm to ~28 nm from an original broad bandwidth. The principle is as follows: when a collimated unpolarized light is incident to the R-PCLC1 cell whose PBG position is between 867 nm and 960 nm, the right-circularly polarized light in the PBG is reflected in the R-PCLC2(S_1′_) direction.

Meanwhile, the PBG position of R-PCLC2 moved to have a PBG position between 802 nm and 895 nm. Therefore, only the right-circularly polarized light at 867~895 nm is reflected by the R-PCLC2 cell. Thus, bandwidth could be decreased from 93 nm to 28 nm. The bandwidth of the bandpass filter system could be reversibly controlled from ~28 nm to 93 nm by moving the PBG position of the R-PCLC2 cell. As the bandwidth decreases, the peak intensity can be reduced. Figure 8b shows bandwidth-reduced bandpass filter spectra from a range of ~425 nm to 895 nm. The transmission does not fall to zero in the outer region of the band in the bandpass filter, which seems to be due to a mismatch in the refractive index at the boundary. Further research seems to be needed on this. In addition, the ripple, which increases the period toward the long wavelength at the band edge, also appears to have occurred due to a refractive index mismatch on the glass substrate.

Figure 9 shows another function of CLC cells as an intensity-variable beam splitter by lateral movement (a) or rotation (b). To study the beam splitter function of PCLC cells, we selected a PBG center position of 633 nm in cells. A 633 nm CW HeNe laser (beam size, FWHM: 0.5 mm) with a max power of 4.745 mW was employed. Cell transmittance increased continuously by up to 90% during lateral movement or rotation. If the PBG center position was changed to 532 nm of the cell and the input laser wavelength was changed to 532 nm, similar behavior happened (see reference paper [14]). This means that this notch filter could work as an intensity-variable beam splitter in the spectral range of ~450 nm to ~950 nm.

### 4.2. Stability for Temporal, Thermal, High-Intensity Laser Input Power

Figure 10a shows PBG data from the newly fabricated R-PCLC1 cell, and Figure 10b shows PBG data of the R-PCLC1 cell measured after heat treatment in a 170 °C oven for one hour. Figure 10c shows PBG data from the R-PCLC1 cell measured about two years after heat treatment in (b). These PBGs were measured as the irradiation beam moved in the thickening direction of the wedge cell (see Figure 2), and the central wavelength position of the PBG was constantly changed from 450 nm to 960 nm. Such good thermal stability and the long-lasting nature of R-PCLC1 were attributed to the polymerization of CLC structures. Other fabricated cells showed similar optical behavior.

Another important factor is stability against high laser power. To study the stability of CLC cells after exposure to high laser power, a 532 nm CW diode laser (Edmund) with a maximum input power of ~143 W/cm^2^, a 446.8 nm CW diode laser (BLM447TB-50FC, Shanghai Laser & Optics Century Co. Ltd., Shanghai, China) with a maximum input power of ~182.4 mW/cm^2^, and a second harmonic generation 532 nm from a Q-switched Nd: YAG laser (pulse width of ~7 ns, 10 Hz) with an input power of ~2.98 MW/cm^2^ were applied to the CLC Notch filter system. A pinhole with 1 mm diameter was used in front of the power meter for excluding the light that passes outside of the PBG center wavelength 532 nm and 446.8 nm, respectively. The Notch filter system consisted of two R-PCLC cells and two L-PCLC cells in a row with a PBG center position of ~532 nm and 446.8 nm. Figure 11 shows the excellent stability of the Notch filter system operated under the high input power of the ND: YAG laser and diode laser for one hour. Data (■), (●), and (▲) are the results of the 532 nm ND: YAG laser, CW 532 nm, and CW 446.8 nm diode laser, respectively. The transmittance at the PBG center (532 nm) was 0.3~0.73% for the ND: YAG laser, 1.25% for the 532 diode laser, and at the PBG center (446.8 nm) was ~0.79% for the diode laser. Interestingly, after long exposure to extremely high laser powers, there was no damage to the filter. These high reflection results might also work as a wavelength variable mirror in VIS and IR wavelength ranges for a high-power laser.

As shown in Figure 5 and Figure 10, when chiral concentration decreased continuously or moved to the thicker direction of the R-PCLC1 cell, the PBG position shifted toward the long wavelength and the bandwidth increased simultaneously. To study variations in the optical constants (i.e., refractive index, pitch, full width at half maximum (FWHM) of the band thickness) of PBGs, three different PBG experimental data with three other wavelength locations were selected and analyzed theoretically. Figure 12 shows three transmittance data of PBGs with center wavelength positions ($\lambda_{B}$) 600 nm, 762 nm, and 964 nm of the R-PCLC1 cell. Solid lines are theoretical fitting data using Berreman’s 4 × 4 matrix method [23,24,25]. Theoretical fitting parameters, refractive index, and thicknesses for these three PBGs are given in Table 2. Additionally, the pitches were obtained by dividing the center wavelength of each PBG by the obtained average refractive index. Other fitting parameters are given for the BK7 substrate with a thickness of 1 mm and a refractive index of 1.5129. In the theoretical fitting, the polyimide (PI) layer was ignored. Due to the AR layer on BK7 in the CLC cell structure (Figure 2), the reflectance at each boundary surface of air/AR layer/BK7 layers was also ignored.

From the fitting results and Figure 2, one of the consequences to understand is that the number of pitches could be changed for different thickness wedge directions. In Figure 1 of reference [26], when one chiral concentration of CLCs is injected into a wedge cell, the pitch length increases by up to 7 nm along the x-axis within the elastic limit of the liquid crystal material to satisfy the boundary condition, resulting in an integer multiple of the half pitch. In this case, the number of pitches is the same between the two Cano lines in the wedge cell. Additionally, as the thickness of the cell increases along the x-axis, the number of half pitches increases by 1 each time it passes through the Cano line. However, in the case of the PCLC cell in this paper, two liquid crystals with two different chiral concentrations, that is, two different pitches, are injected into each end of the wedge cell at different thicknesses, respectively. The chiral molecular concentration gradient is formed along the x-direction over time while satisfying the boundary condition. Therefore, the number of pitches could be changed throughout the wedge cell [13,15,26]. Thus, there is no Cano line because the pitch changes continuously (Figure 3b). As shown in Figure 12, all three PBG data were somewhat consistent with theoretical fit curve data. There were some differences between the measured data and theoretically fitted curves near band edges. Experimentally measured values were gently decreased. However, in theoretically fitted curves, large oscillations occurred near the band edge. In theory, molecules were considered to be aligned with the perfect helical structure by cooperation between the anchoring force of the alignment layer and helical rotary powers of chiral molecules. However, in an actual CLC cell with a large thickness, it was hard to form a perfect helical structure due to weak cooperation in the bulk area far from the alignment layer because the helical structure of molecules in the CLC cell originated from the cooperation of the anchoring energy between CLC molecules, the alignment layer (PI), and the helical rotary power of chiral molecules [15,23,26]. When applying CLC cells to real devices, a gentle reduction near the edge of the band might be advantageous.

## 5. Conclusions

Using polymerized cholesteric liquid crystal cells, we realized a practically applicable multifunctional optical device with continuous wavelength tunability over a 500 nm spectral range. It has multiple functions. For example, it can be used as a circular polarizer generating an extremely high degree of circularly polarized light with a dissymmetry factor of |g|=1.85~2.00. It can also be used as an optical notch filter, a bandwidth-variable (from ~28 nm to ~93 nm) bandpass filter, a mirror, and intensity-variable beam splitters. Furthermore, owing to the polymerization of cholesteric liquid crystal cells, it has high stability for a long time (~2 years), after a high-temperature exposure (170 °C for 1 h), and after an extremely high laser beam intensity exposure (~143 W/cm^2^ of CW 532 nm diode laser and ~2.98 MW/cm^2^ of Nd: YAG pulse laser operation for two hours, respectively). The characteristics of optical devices were significantly improved by introducing an anti-reflection layer to the device. The optical properties of the refractive index, pitch, and FWHM of the PBG and the thickness of the polymerized CLCs were theoretically analyzed using Berreman’s 4 × 4 matrix method. If PCLC materials with much smaller ∆n can be synthesized and applied to this kind of filter device, filter resolution and bandwidth variation can be increased, and application efficiency will be maximized. Such devices could highly increase research efficiency in various spectroscopic works. They might be applied to many instruments using VIS and NIR wavelengths.

## Figures and Tables

**Figure 1 polymers-13-03720-f001:**
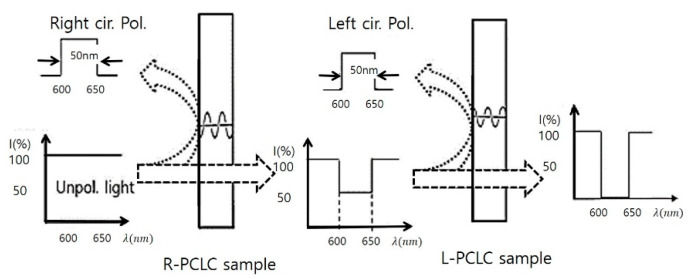
Principle of a circular polarizer by one R-PCLC (or L-PCLC) and a notch filter by two R- PCLCs. A notch filter works for unpolarized incident light, right-circularly polarized light in a photonic band is reflected at the R-PCLC cell, and then left-circularly polarized light in the same photonic band is reflected at the L-PCLC cell, in turn.

**Figure 2 polymers-13-03720-f002:**
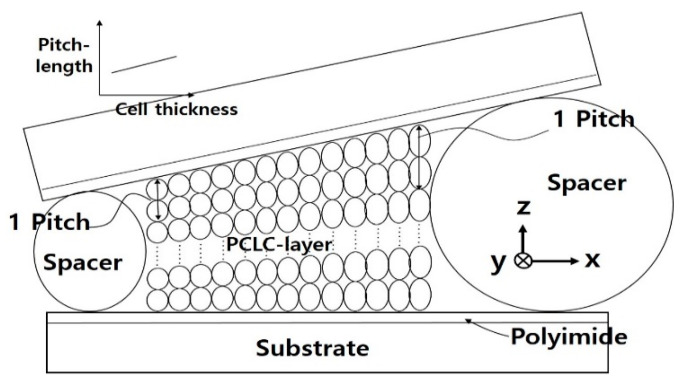
Schematic diagram of the wedge PCLC cell with continuous pitch gradient.

**Figure 3 polymers-13-03720-f003:**
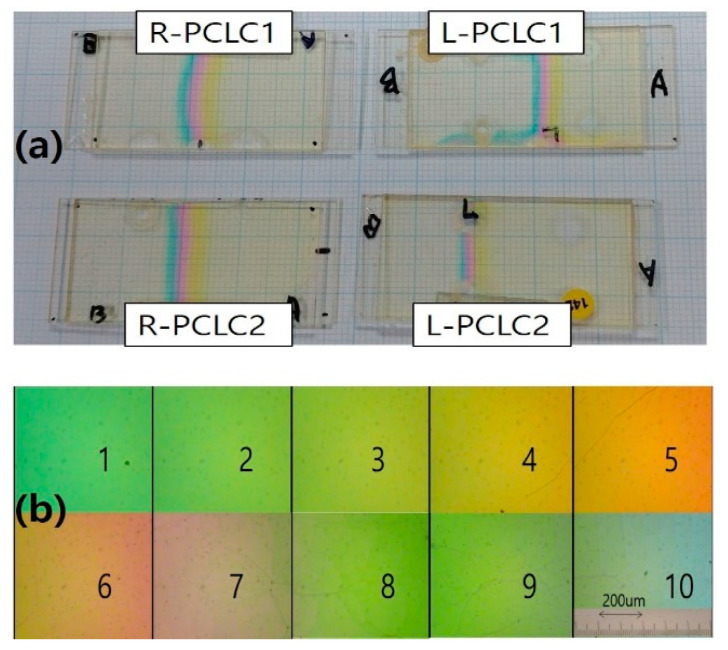
(**a**) Photographs of well-developed wedge cells of R-PCLC1, R-PCLC2, L-PCLC1, and L-PCLC2 placed on a plotting paper, and (**b**) Juxtaposition of ten polarized microscope photographs at different spatial positions of the R-PCLC1 cell, respectively.

**Figure 4 polymers-13-03720-f004:**
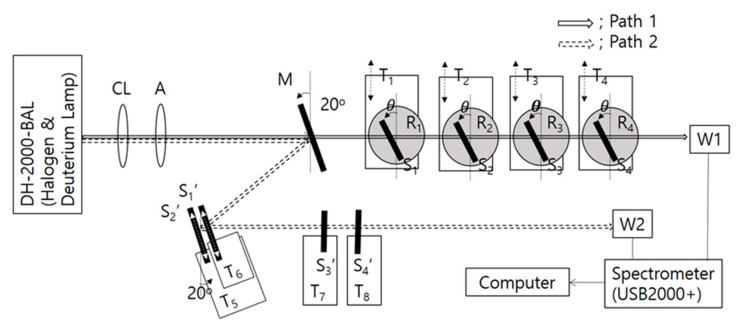
Experimental setup. M, Ag mirror; CL, lens for a collimated beam; A, aperture; R1~R4, rotator; T1~T8, translation stages; $S_{1},\;S_{2},\;S_{3},\;S_{4},\;S_{1}^{\prime },\;S_{2}^{\prime },\;S_{3}^{\prime },\;\mathrm{and}\;S_{4}^{\prime },$ R- or L-PCLC cells; and W1 and W2, waveguides of 400 μm diameter. Depending on the purpose of the device, some of M, R-, or L-PCLC cells could be removed from the setup. To study the function of the circular polarizer and notch filter, Path 1 was used. To study the bandpass filter, Path 2 was used.

**Figure 5 polymers-13-03720-f005:**
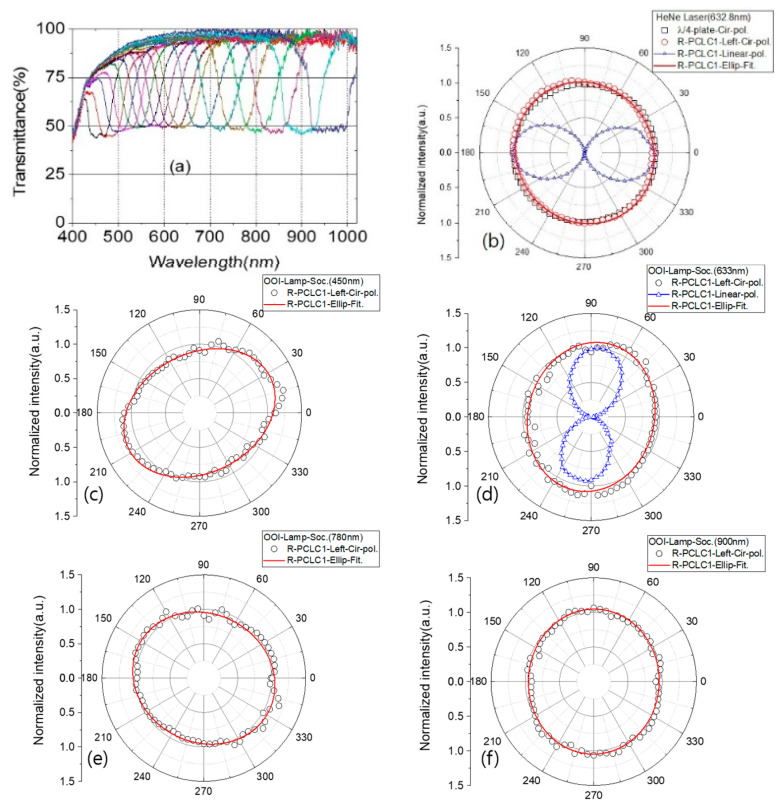
(**a**) Continuous PBG transmission spectra of the R-PCLC1 cell are made by moving it laterally for vertically incident beams. (**b**) Polar plot by using HeNe laser (632.8 nm); (□), a left-circularly polarized light made by a linear polarizer and a λ4 plate of 632.8 nm; (o), a left-circularly polarized light passed through the R-PCLC1 cell with a PBG center wavelength at 633 nm; (-) a fitting curve by an elliptic equation; (△), a linearly polarized light intensity that is converted from the data (o) by a λ4 plate (632.8 nm). Polar plots by using the halogen and deuterium lamp; the lights passed through the R-PCLC1 cell with a PBG center wavelength at 450 nm (**c**), 633 nm (**d**), 780 nm (**e**), and 900 nm (**f**), respectively. The solid red lines are fitting curves by an elliptic equation. The data (△) in (**d**) are a linearly polarized light intensity that is converted from the data (o) of (**d**) by a λ4 plate (632.8 nm).

**Figure 6 polymers-13-03720-f006:**
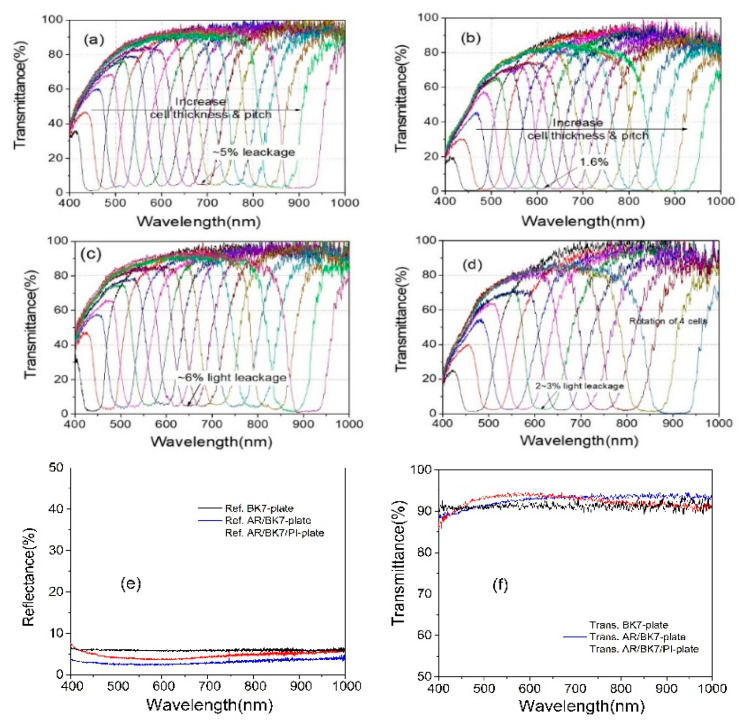
Spectra of continuous tuning notch filter system in Path 1 of Figure 4 (**a**) via lateral movement of one notch filter system and (**b**) via lateral movement of two notch filter systems. Notch filter spectra (**c**) via rotation of one notch filter system or (**d**) via rotation of two notch filter systems. Data are shown in a low density to distinguish one by one easily. (**e**) Reflectance of BK7 plate, AR/BK7 plate, and AR/BK7/PI plate. (**f**) Transmittance of BK7 plate, AR/BK7 plate, and AR/BK7/PI plate, respectively.

**Figure 7 polymers-13-03720-f007:**
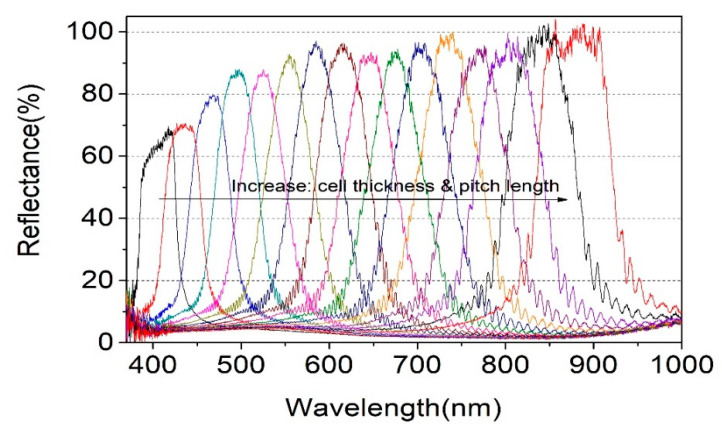
Spectra from the continuous tunable bandpass filter system. Data are shown in a low density to distinguish one by one easily.

**Figure 8 polymers-13-03720-f008:**
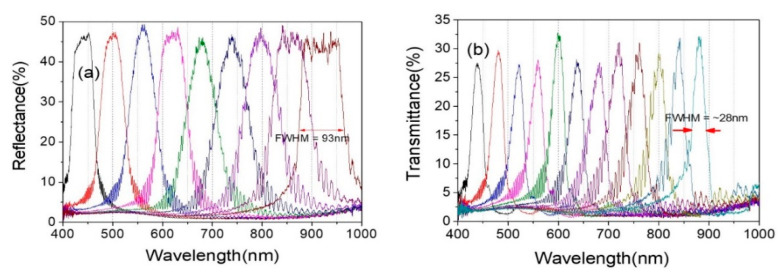
Original PBG bandwidth of the R-PCLC1 cell (**a**) and reduced PBG bandwidth spectra (**b**). Data are shown in a low density to distinguish one by one easily.

**Figure 9 polymers-13-03720-f009:**
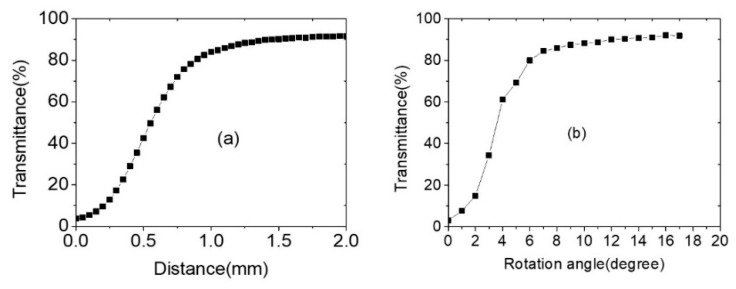
The transmittance of R-PCLC cell via lateral movement (**a**) and rotation movement (**b**) for incident 632.8 nm HeNe laser at 4.67 mW/cm^2^.

**Figure 10 polymers-13-03720-f010:**
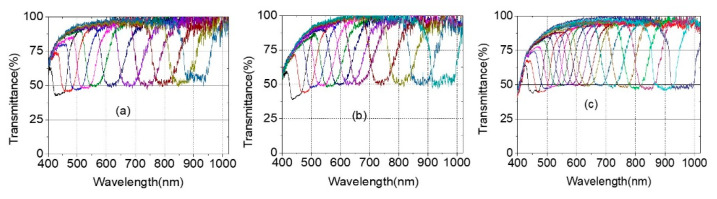
Continuous PBG transmission spectra of R-PCLC1 wedge cell for vertical incident beams in Path 1 of Figure 4. (**a**) PBG data from the newly fabricated cell, (**b**) PBG data after thermal treatment in a 170 °C oven for 1 h, (**c**) PBG data after about two years have elapsed since the thermal treatment in (**b**). Although PBGs could be measured at a high density with a fine spatial movement, data are shown in a low density to distinguish one by one easily.

**Figure 11 polymers-13-03720-f011:**
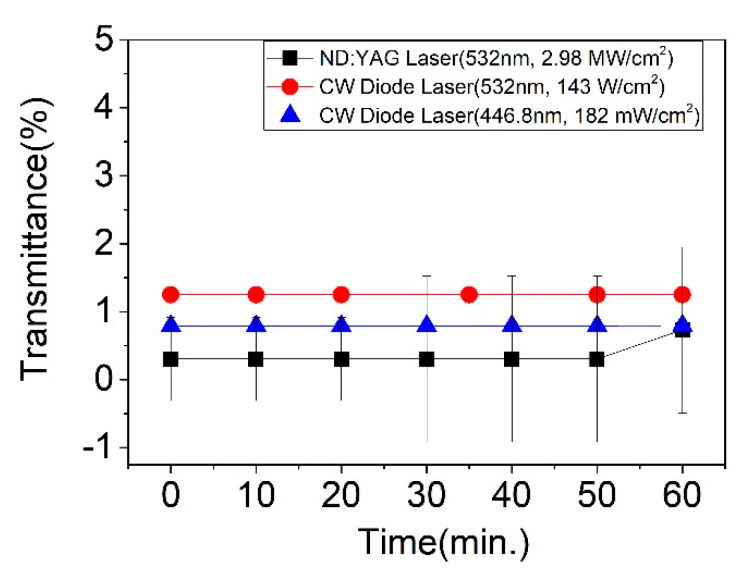
Stability and transmittance of filters via a high-pulse ND: YAG laser (■) at 532 nm and a CW diode laser at 532 nm (●) and a CW diode laser at 446.8 nm (▲) for one hour.

**Figure 12 polymers-13-03720-f012:**
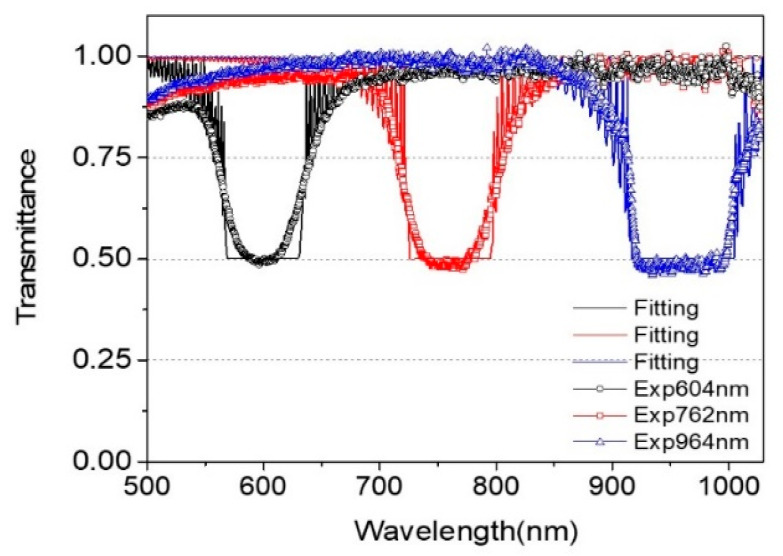
Experimentally measured three PBG data and their theoretical fitting data (solid lines) of the R-PCLC1 cell with three PBG center wavelength positions: λ_B_= 600 nm (o), 762 nm (□), and 964 nm (△).

**Table 1 polymers-13-03720-t001:** The length of semi-major axis a, the length of semi-minor axis b, and the ellipticity (b/a) were obtained by fitting the circularly polarized light data in Figure 5 to the elliptic equation.

Light Source (λ)	Sample	Semi-Major Axis (a)	Semi-Minor Axis (b)	Ellipticity (b/a)
HeNe Laser (633 nm)	632.8 nm-(λ4) plate	1.015	0.985	0.970
HeNe Laser (633 nm)	R-PCLC1	1.033	0.969	0.938
OOI Lamp (450 nm)	R-PCLC1	1.163	0.868	0.746
OOI Lamp (633 nm)	R-PCLC1	1.090	0.922	0.846
OOI Lamp (780 nm)	R-PCLC1	1.072	0.934	0.871
OOI Lamp (900 nm)	R-PCLC1	1.048	0.956	0.912

**Table 2 polymers-13-03720-t002:** Optical constants of three PBGs of the R-PCLC1 cell with the center wavelength position of 600 nm, 762 nm, and 964 nm.

**λ_B_**	600 nm	762 nm	964 nm
**n_e_**	1.705	1.675	1.675
**n_0_**	1.537	1.528	1.528
**Pitch**	370 nm	476 nm	602 nm
**FWHM**	78 nm	89 nm	103 nm
**Thickness**	23.0 μm	27.5 μm	30.0 μm

## Data Availability

Not applicable.

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
