# Peer review of "Multifunctional Optical Device with a Continuous Tunability over 500 nm Spectral Range Using Polymerized Cholesteric Liquid Crystals"

_polymers, 2021, doi:10.3390/polym13213720_

Round 1

Reviewer 1 Report

Paper ID: polymers-1418954

Title: "Multifunctional optical device with a continuous tunability over 500 nm spectral range using polymerized cholesteric liquid crystals

Authors:

Mi-Yun Jeong*, Hyeon-Jong Choi, Keumcheol Kwak, and Younghun Yu

Reviewer's comments:

  1. The first phrase of the Abstract, "We report the polymerization makes a robust, practically applicable multifunctional opti cal device with a continuous wavelength tunable over 500 nm spectral range using UV-polymerizable cholesteric liquid crystal (CLC)",

marked in red by the authors, does not make any sense and it should be corrected.

  1. In section3, row 150, the UV-cured CLC is mentioned, BUT there is no information about the starting liquid crystal. What was its formula, is it commercial, is it a mixture?? How was it obtained? More details should be given here.

  1. Regarding the References: ALL references should be written in the format of the journal, that is Full Family Name Of All Authors, innitials of the name, etc.,.... and the doi of the paper, where it is available. All references should be written accordingly.

The paper can be published only after the above mentioned corrections are operated.

Author Response

Reviewer's comments:

  1. The first phrase of the Abstract, "We report the polymerization makes a robust, practically applicable multifunctional optical device with a continuous wavelength tunable over 500 nm spectral range using UV-polymerizable cholesteric liquid crystal (CLC)",

marked in red by the authors, does not make any sense and it should be corrected.

          Answer; The first phrase of the abstract is corrected.

2. In section3, row 150, the UV-cured CLC is mentioned, BUT there is no information about the starting liquid crystal. What was its formula, is it commercial, is it a mixture?? How was it obtained? More details should be given here.

        Answer; All CLC materials are commercial. As mentioned in Section "3. Fabrication of PCLC cells", some of them are mixtures. Some purchased directly from Merck. And some of them were obtained from other laboratories. The author could never get any more information about the material formula by their security policy. However, the principles applied to this paper do not seem to be limited to the materials we used.

3. Regarding the References: ALL references should be written in the format of the journal, that is Full Family Name Of All Authors, innitials of the name, etc.,.... and the doi of the paper, where it is available. All references should be written accordingly.

        Answer; All references have been modified in the form of journals.

  • Thank you for your careful review.

      Thanks to this, the shortcomings of the thesis have been supplemented a lot.

Reviewer 2 Report

The article "Multifunctional optical device with a continuous tunability over 500 nm spectral range using polymerized cholesteric liquid crystals" by Mi-Yun Jeong and co-authors presents the investigation of wedge-cell of polymerized CLC as multifunctional optical device (circular polariser; tunable notch filters, bandpass filters, mirrors, and intensity variable beam splitters. The stability of the transmission spectra and the laser irradiation transmittance on exposure to high temperature or high-power laser radiation has been studied. The analysis of obtained results in the revised version of the manuscript has undoubtedly improved. However, I still have a few comments on the manuscript.

1. The difference between refractive indices of the PI layer and PCLC layer or PI/BK7 is used to explain such features of the spectra as:

- 5% light leakage in the notch filter system (Lines 302-306);

- the decreasing from ~90% to 20% of band's transmittance of the outer region of the notch filter set (Lines 329-337);

- the decreasing from ~90% to 60% of the bandpass filter system (Lines 349-352);

- nonzero transmittance and ripple in the outer region of the band in the bandpass filter (Lines 373-377).

This argument/assumption seems unfounded to me. I think that in the manuscript should be added References where these effects were studied, or calculations of influencing the PI layer on the transmission/reflection spectra (for example, using Berreman's 4x4 matrix method).

2. It is not clear from the text why a 532 nm laser was used to test the system's stability against high-power radiation. Will the system remain stable if the laser light wavelength is changed? For example, at the light with a wavelength of 450 nm, which, apparently, is absorbed by the material from which the PCLC has been made.

3. How were the data in Table 2 obtained? For the given in Table 2 values of the thicknesses d and the CLC pitch P, the number of CLC layers is approximately N=61.7 at d=23 μm, N=57.8 at d=27.5 μm, and N=49.8 at d=30 μm. In the model shown in Figure 2 the number of CLC layers must be the same. What is the structure of cholesteric formed in CLC cells?

4. The text contains many misprints.

Author Response

  1. The difference between refractive indices of the PI layer and PCLC layer or PI/BK7 is used to explain such features of the spectra as:

- 5% light leakage in the notch filter system (Lines 302-306);

- the decreasing from ~90% to 20% of band's transmittance of the outer region of the notch filter set (Lines 329-337);

- the decreasing from ~90% to 60% of the bandpass filter system (Lines 349-352);

- nonzero transmittance and ripple in the outer region of the band in the bandpass filter (Lines 373-377).

This argument/assumption seems unfounded to me. I think that in the manuscript should be added References where these effects were studied, or calculations of influencing the PI layer on the transmission/reflection spectra (for example, using Berreman's 4x4 matrix method).

Answer; To supplement the interpretation, additional experiments were conducted and new results are added in Figure 6;  Lines, 353~367

From the Fig. 6 (b) and (d) results, in order to examine the effect of the PI layer on the serious decrease in transmittance from 600 nm to 400 nm range, the reflectance and transmittance of the substrate used to manufacture the PCLC cell were investigated. Figure 6(e) shows reflectance of BK7- plate, AR/BK7-plate, and AR/BK7/PI-plate. And Figure 6 (f) shows transmittance of BK7- plate, AR/BK7-plate, and AR/BK7/PI-plate, respectively. In the case of AR/BK7 substrates, AR coating on one side of the BK7 substrate reduces (increases) reflectance (transmittance) uniformly in the wavelength region of 400 nm to 1000 nm. However, for AR/BK7/PI-plate, the reflectance (transmittance) rate increases (decreases) again, and this effect increases, especially as it moves from 600 nm to 400 nm region. It seems to be the effect of refractive index mismatch and absorption at the interface between the BK7/PI layers in this wavelength region. Therefore, as the number of PCLC cells used increases, the number of substrates increases, and the transmission results of Figs. 6(b) and 6(d) appear, and this part needs to be improved further in the future.

  1. It is not clear from the text why a 532 nm laser was used to test the system's stability against high-power radiation. Will the system remain stable if the laser light wavelength is changed? For example, at the light with a wavelength of 450 nm, which, apparently, is absorbed by the material from which the PCLC has been made.

Answer; Lines, 448~461

 Additionally, with the laser that we can get so far, we measured the stability for blue diode CW laser at wavelength 446.8nm (BLM447TB-50FC, BE30S09300, Shanghai Laser & Optics Century Co. Ltd.). The result is added in figure 11 and explained about that.

  1. How were the data in Table 2obtained? For the given in Table 2 values of the thicknesses d and the CLC pitch P, the number of CLC layers is approximately N=7 at d=23 μm, N=57.8 at d=27.5 μm, and N=49.8 at d=30 μm. In the model shown in Figure 2 the number of CLC layers must be the same. What is the structure of cholesteric formed in CLC cells?

Answer; In Table 2, the refractive index and thickness were obtained by computer simulation using the data in Figure 12. And the pitches were obtained by dividing the center wavelength of each PBG by the obtained average refractive index. (Refer to eq,  , line 97)

Lines, 494 ~ 507

 From the fitting results and the Figure 2, one of the consequences to understand is that the number of pitches could be changed for different thickness wedge direction. For a more detailed explanation, let's first consider Figure 1 of Reference [26]. When one chiral concentration of CLC is injected into a wedge cell, the pitch length increases by up to 7 nm within the elastic limit of the liquid crystal to satisfy the boundary condition, resulting in an integer multiple of the half pitch. In this case, the number of pitches is the same between the two cano lines in the wedge cell. And as the thickness of the cell increases along the x-axis, the number of half pitches increases by 1 each time it passes through the canoe line. After understanding this CLC cell with one fixed chiral concentration situation clearly, consider the current situation of this paper. In the case of the PCLC cell in this paper, two liquid crystals with two different chiral concentrations, that is, two different pitches, are injected into each end of the wedge cell at different thicknesses, respectively. The chiral molecular concentration gradient is formed along the x-direction over time while satisfying the boundary condition. Therefore, the number of pitches could be changed throughout the wedge cell. Please read the following references to understand this in detail. First, read reference 26 and then read 13 and 14 in the References. Therefore, the structure of CLC formed in cells is shown in Figure 2.

Ref. 13. M.-Y. Jeong, J. W. Wu. Continuous spatial tuning of laser emissions with tuning resolution less than 1 nm in a wedge cell of dye-doped cholesteric liquid crystals. Opt. Express 2010, 18, doi:10.1364/OE.18.024221. 

Ref. 15 M.-Y. Jeong, J.W.Wu. Continuous Spatial Tuning of Laser Emissions in a Full Visible Spectral Range. Int. J. Mol. Sci. 2011, 12, doi:10.3390/ijms12032007.

  Ref. 26. Mi-Yun Jeong, H.Choi, J. W. Wu. Spatial tuning of laser emission in a dye-doped cholesteric liquid crystal wedge cell. Applied Physics Letters 2008, 92, doi:10.1063/1.2841820, doi: 10.1063/1.2841820.

  1. The text contains many misprints.

Answer; Misprints in the text were corrected as much as possible.

  • Thank you for your careful review.

Thanks to this, the shortcomings of the thesis have been supplemented a lot.

This manuscript is a resubmission of an earlier submission. The following is a list of the peer review reports and author responses from that submission.

Round 1

Reviewer 1 Report

The article "Multifunctional optical device with a continuous tunability over 500 nm spectral range using polymerized cholesteric liquid crystals" by Mi-Yun Jeong and co-authors presents the investigation of wedge-cell of polymerized CLC as multifunctional optical device (circular polariser; tunable notch filters, bandpass filters, mirrors, and intensity variable beam splitters. The stability of the transmission spectra and the laser irradiation transmittance on exposure to high temperature or high-power laser radiation has been studied.

In my opinion, research is not well developed, and the manuscript cannot be recommended for publication in this form.

A few comments on the manuscript are as follows:

1. The used degree of circular polarization (dissymmetry factor?) g characterizes only the intensity of the right circularly polarized radiation transmitted (reflected) through the sample. In the present form, g does not characterize the polarization of the transmitted light (see, for example, book “Physical Optics and Light Measurements” Edited by Daniel Malacara or paper Sascha Trippe // Journal of The Korean Astronomical Society, Vol. 47, No.1, P. 15.) and, accordingly, cannot be used to determine the quality of a circular polarizer.

2. The CLC cell used is suggested as the circular polarizer in a wide wavelength range. Nevertheless, the data on the polarization ellipticity is given only for 633 nm light wavelength (Figure 5b), at which T = 50% (Figure 5a). What is the polarization ellipticity for other light wavelengths, especially for 450 nm and 960 nm, at which T is not equal to 50% (Figure 5a)?

3. Figure 6 shows the transmission spectra of a system of two and four CLC cells (the notch filter systems). The dependence of FWHM of PBG on the filter wavelength has been analyzed only for the two-CLC-cells system (Figure 6 a,c). However, it can be seen from Figure 6 that the FWHM of PBG of the four-CLC-cells system is greater than in the case of the two-CLC-cells system. This experimental fact is not discussed or analyzed in the manuscript.

4. The explanation for the presence of light leakage seems to be incorrect (Lines 273-299). In the case of a single R-PCLC cell (L-PCLC cell), the left (right) circularly polarized light leakage is approximately equal to 0.5% (see Line 236). Thus, for the system of R-PCLC and R-PCLC cells, the unpolarized light leakage should also be about 0.5%.

5. For bandpass filters (Figure 7), out-of-band transmittance is an important parameter. However, these values are not indicated in the manuscript and are not analyzed.

6. In the study of the temporal and thermal stability of the system, it was shown that it remains possible to tune the PBG position of the CLC cell (Figure 10). It is not clear from the text whether the dependence of the spectral position of PBG on the x-coordinate is preserved. Or after thermal action (over time), will the minimum transmission at some wavelength (for example, 500 nm) correspond to different x-coordinate values?

7. It is not clear from the text why a 532 nm laser was used to test the system's stability against high-power radiation. Will the system remain stable if the laser light wavelength is changed? For example, at the light with a wavelength of 450 nm, which, apparently, is absorbed by the material from which the PCLC has been made.

8. How were the data in Table 1 obtained? For the given in Table 1 values of the thicknesses d and the CLC pitch P, the number of CLC layers is approximately N=61.7 at d=23 μm, N=57.8 at d=27.5 μm, and N=49.8 at d=30 μm. In the model shown in Figure 2 the number of CLC layers must be the same. What is the structure of cholesteric formed in CLC cells?

9. The text contains many misprints. For example, on Lines 99 (“p” and “P”), 163 (“RM-xx”), 326 (“….than those of Fig. 7.”), and References.

Reviewer 2 Report

Jeong et al. report a system for fabricating a multifunctional optical device both with high thermal stability and optical tunability over 500 nm spectral range based on the cholesteric liquid crystal (CLC) polymers. As the author described that CLC intrinsically has low thermal stability due to the unique optical property arising from its helical molecular orientation should change under excessive heating, the polymerization of CLC enables to enhance the thermal stability due to the fixation of molecular orientation by polymer network. In addition, the polymerized CLC have kept its original optical property of CLCs.

Though we have not found any technical problems in the paper and the design can be useful for some practical applications, we think the underlying strategy demonstrated in the study is relatively trivial and there are some fundamental flaws as described below.

  1. In the introduction section and Figure 2, the author described that the wedge cell is important to generate pitch gradient of the wedge CLC, which is a main reason of broadband optical functions. However, in the wedge cell, the CLC can form a pitch gradient upto half pitch of an initial helical structure. This means that, if one uses the CLC with 300 nm pitch at initial state, the wedge cell can form the gradient between 300 nm to 450 nm theoretically.
    After carefully reading the manuscript, the author expressed that the reason of such wide gradient of helical pitch is concentration gradient of chiral dopant that induce a helical structure. This gradient is formed by injecting two different mixtures with high concentration and low concentration of chiral dopant. This is very easy to understand and has already well-known effect in principle. For example, Broer et al. have developed novel procedure to generate a pitch gradient for realizing wide-band reflection in 1995 (Nature). Perhaps, the present report by Jeong et al. have some novelty to generate a pitch gradient in the in-plane direction of a wedge cell, they have already published the experimental result in 2018 (Applied Optics).
  2. Considering previous reports of similar studies, as the author also cited, the present work might have a progress on the increment of thermal stability and the fixation of molecular diffusivity of CLC systems by polymerizing them. However, to my best knowledge, it is very typical that the polymerization of LC medium enhances the thermal stability of molecular orientation as well as optical properties and fixes the molecular position. In addition, the author did not show chemical structures used in this study, the polymer molecular weight, and whether the polymer is crosslinked or not. Some of such information are required to ensure why the stability are established.

Unfortunately, we think the present work lacks sufficient novelty to be published.

Reviewer 3 Report

Paper: Multifunctional optical device with a continuous tunability over 500 nm spectral range using polymer-2 ized cholesteric liquid crystals

Manuscript ID: polymers-1350680

Special Issue: Polymer Composites and Films in Display Devices

Reviewer's observations:

  1. The introduction, starting with r. 37, should be re-written: a clear explanation of what CLC are (chiral nematics) should be given. The phrase: "CLC has various possibilities" (r.39) does not make any sense

The authors might as well cite the papers: Gilli, J. M., Thiberge, S., & Manaila-Maximean, D. (2004). New aspect of the voltage/confinement ratio phase diagram for a confined homeotropic cholesteric. Molecular Crystals and Liquid Crystals, 417(1), 207-213. doi: 10.1080/15421400490478858 and

Bărar, A., Dănilă, O., Mănăilă-Maximean, D., & Loiko, V. A. (2019, September). Active spectral absorption control in a tunable liquid crystal/metamaterial structure by polarization plane rotation. In International Conference on Nanotechnologies and Biomedical Engineering (pp. 299-303). Springer, Cham. •DOI: 10.1007/978-3-030-31866-6_58

  1. r.59, the idea in the phrase: "Despite the superior performance of such CLC devices, CLC cells were made by general CLCs" is not clearly explained.
  2. Figure 1 is identical with Fig. 1 in the paper: Jeong, M. Y., & Kwak, K. (2018). Continuously tunable and bandwidth variable optical notch/band-pass filters over 500 nm spectral range using cholesteric liquid crystals. IEEE Photonics Journal, 11(1), 1-11.

DOI: 10.1109/JPHOT.2018.2884965

  1. Figure 2 is identical with Fig. 2 in the paper: Jeong, M. Y., & Kwak, K. (2018). Continuously tunable and bandwidth variable optical notch/band-pass filters over 500 nm spectral range using cholesteric liquid crystals. IEEE Photonics Journal, 11(1), 1-11.

(at least). DOI: 10.1109/JPHOT.2018.2884965. The editor should check if they have the copyright or the authors should replace the figures.

  1. In "3. Fabrication of PCLC cells" more details about the sample preparation should be given: what are the respective concentrations of the components in the mixtures before polymerization. Details about their molecular mass. Why the need of polymerization, what is the difference in the structure of the material in the non-polymerized and in the polymerized samples.
  2. Fig. 3(a) should be larger.
  3. The authors should stress the advancement presented in this paper as compared to their previous devices.

Reviewer's recommendation: major revision